# Block Copolyesters Containing 2,5-Furan and *trans*-1,4-Cyclohexane Subunits with Outstanding Gas Barrier Properties

**DOI:** 10.3390/ijms20092187

**Published:** 2019-05-02

**Authors:** Giulia Guidotti, Laura Genovese, Michelina Soccio, Matteo Gigli, Andrea Munari, Valentina Siracusa, Nadia Lotti

**Affiliations:** 1Department of Civil, Chemical, Environmental and Materials Engineering, University of Bologna, Via Terracini 28, 40131 Bologna, Italy; giulia.guidotti9@unibo.it (G.G.); lauragenovese87@gmail.com (L.G.); m.soccio@unibo.it (M.S.); andrea.munari@unibo.it (A.M.); 2Department of Chemical Science and Technologies, University of Roma Tor Vergata, Via della Ricerca Scientifica 1, 00133 Roma, Italy; matteo.gigli@uniroma2.it; 3Department of Chemical Science, University of Catania, Viale A. Doria 6, 95125 Catania, Italy

**Keywords:** biopolyesters, furanoate-based polymers, poly(propylene furanoate), block copolymers, gas barrier properties, mesogenic groups

## Abstract

Biopolymers are gaining increasing importance as substitutes for plastics derived from fossil fuels, especially for packaging applications. In particular, furanoate-based polyesters appear as the most credible alternative due to their intriguing physic/mechanical and gas barrier properties. In this study, block copolyesters containing 2,5-furan and *trans*-1,4-cyclohexane moieties were synthesized by reactive blending, starting from the two parent homopolymers: poly(propylene furanoate) (PPF) and poly(propylene cyclohexanedicarboxylate) (PPCE). The whole range of molecular architectures, from long block to random copolymer with a fixed molar composition (1:1 of the two repeating units) was considered. Molecular, thermal, tensile, and gas barrier properties of the prepared materials were investigated and correlated to the copolymer structure. A strict dependence of the functional properties on the copolymers’ block length was found. In particular, short block copolymers, thanks to the introduction of more flexible cyclohexane-containing co-units, displayed high elongation at break and low elastic modulus, thus overcoming PPF’s intrinsic rigidity. Furthermore, the exceptionally low gas permeabilities of PPF were further improved due to the concomitant action of the two rings, both capable of acting as mesogenic groups in the presence of flexible aliphatic units, and thus responsible for the formation of 1D/2D ordered domains, which in turn impart outstanding barrier properties.

## 1. Introduction

At present, polymers and plastics are mostly obtained from fossil sources. The decline of petroleum reserves, the fluctuating price of petroleum based products, and the stringent environmental regulations due to severe environmental pollution and waste management issues have prompted the use of chemicals derived from renewable resources in both scientific and industrial communities [1,2,3]. A shift towards the use of biomass for the production of polymers would indeed not only reduce the contribution to the greenhouse effect, but also preserve mineral resources for future generations. It is important to note that packaging represents the largest market segment in the plastic industry. More than a third of all plastics yearly produced are converted into packaging (approximately 100 million tons worldwide). In Western industrial countries, 50 percent of all goods are packaged in plastics [4]. Without the various available packaging solutions, many sensitive goods would waste away or get damaged en route to the customer. Recently, also because of the growing concern and awareness on sustainability, the development and use of bioplastics for food packaging has become of great interest from an industrial point of view.

Among possible alternatives, polyesters containing furan moieties are an emerging and very promising class of biobased materials. Indeed, 2,5-furandicarboxylic acid (FDCA) can be derived from 5-Hydroxymethylfurfural (HMF), in turn obtained from sugars and polysaccharides [5]. The latest developments in FDCA synthesis [6,7,8] have led to the production of numerous FDCA-based polyesters such as poly(ethylene 2,5-furandicarboxylate) (PEF) [9,10,11,12,13], poly(propylene 2,5-furandicarboxylate) (PPF) [14,15,16], poly(butylene 2,5-furandicarboxylate) (PBF) [17,18,19,20,21], to cite the most important examples, and others including some FDCA-based copolyesters and nanocomposites [22,23,24,25,26,27,28,29,30,31,32,33,34,35,36,37,38,39]. Poly(ethylene 2,5-furandicarboxylate) (PEF) represents a very promising solution, particularly as a substitute of poly(ethylene terephthalate) (PET), owing to very similar physic/mechanical properties. Furthermore, PEF displays considerably enhanced gas barrier properties as compared to PET. Specifically, a 11×, 19×, and 2.1× lowering of O_2_, CO_2_, and water permeability, respectively, has been observed for amorphous PEF with respect to amorphous PET [40,41,42]. Barrier properties of PPF [14,16], PBF [20,34,36,37], poly(pentamethylene furanoate) (PPentF) [43], and poly(neopenthyl glycol furanoate) (PNF) [44] have also been recently investigated in order to evaluate their potential for food packaging applications.

However, homopolymers do not always fulfill all the requirements for a specific application. In this respect, a very useful technique to modulate the material characteristics in order to achieve the desired combination of properties is represented by copolymerization. Copolymerization allows for the creation of completely new classes of materials, whose behavior is strictly correlated to the composition and molecular architecture of the comonomeric units present in the polymer structure. Over the past several years, our group has intensively focused the research activity on the synthesis and characterization of fully biodegradable and/or biobased aliphatic copolyesters from *trans*-1,4-cyclohexanedicarboxylic acid for environmental as well as biomedical applications [45,46,47,48,49,50,51,52,53,54,55]. Such polymers proved to be particularly interesting for food packaging applications [45,49,51,52,53,54]. Recently, we focused our attention on the fully-aliphatic biobased poly(propylene cyclohexanedicarboxylate) (PPCE), whose thermal and mechanical properties were deeply investigated, together with its compostability [55].

In this framework, fully biobased multiblock copolymers of PPF and PPCE with fixed composition and different block length (PPFPPCE-t, with t equal to mixing minutes) were prepared through reactive blending, a simple, solvent-free and easily scalable synthetic route, with the aim of reducing the intrinsic PPF rigidity while preserving its outstanding gas barrier behavior, thus widening its range of applications. The thermal and mechanical properties of the synthesized copolymers were investigated and correlated to the molecular architecture. Last, but not least, to evaluate the potential of these new polyesters for flexible food packaging, the barrier performances to different gases and at various temperatures were tested.

## 2. Results and Discussion

### 2.1. Polymer Synthesis and Molecular and Thermal Characterization

The copolymer synthesis (chemical structure in Figure 1) was optimized by varying the reaction temperatures in the range 210–260 °C. At 210 °C the transesterification reactions did not appreciably occur over a reaction period of 8 h, in the range 220–230 °C the reaction kinetic rate was very low and at 250 °C, the prevalence of chain scission reactions prevented the obtainment of copolymers with high molecular weight. Therefore, 240 °C was chosen as the most suitable option.

Molecular characterization data are reported in Table 1. All the samples display relatively high and similar molecular weight, proving that no significant thermal degradation occurred during the reactive blending. Moreover, the higher the mixing time, the higher the molecular weight of the multiblock copolyesters. This effect, already reported in the literature, is due to the prevalence of transesterification reactions over chain scissions [50].

Figure 2 reports the ^1^H-NMR spectrum of PPFPPCE-40 with the corresponding resonance assignments. Besides confirming the expected structure, it can be observed that, as suggested by the low intensity of d_cis_ protons with respect to the d_trans_ ones, the *cis*/*trans* ratio of the cyclohexane moiety did not vary with respect to the feed (3%), regardless of the high temperature reached during the polymerization process. The copolymer composition was determined from the relative areas of the resonance peak of the aromatic protons of the furan ring located at 7.38 ppm and of the signal at 1.48 ppm of the f protons of the cyclohexane moiety (Figure 2). The actual composition was close to the feed in all cases (Table 1). In addition, NMR spectroscopy permitted us to follow the structural changes occurring during reactive blending by monitoring the evolutions of the peaks located in the region between 4.6 and 4.2 ppm (Figure 2, bottom), where the propanediol protons of the –OCH_2_− group are located. In this region, besides the triplets at 4.58 and 4.24 ppm, corresponding to the F–P–F (b protons) and CE–P–CE (g protons) triads, with the increase of the mixing time, two additional triplets arose due to the F–P–CE and CE–P–F (i and j protons) triads that formed as a consequence of transesterification.

The degree of randomness ***b*** was calculated from the relative intensity of the signals corresponding to b, g, i, and j protons. ***b*** is equal to 1 for random copolymers, 2 for alternate copolymers, 0 for a mixture of two homopolymers, and 0 < ***b*** < 1 for block copolymers. The degree of randomness was calculated according to Equation (1):***b*** = P_F-CE_ + P_CE-F_(1)
where P_F-CE_ and P_CE-F_ are the probability of finding an F unit next to a CE one and the probability of finding a CE unit next to an F one, respectively. In turn, P_F-CE_ and P_CE-F_ can be expressed as follows (Equation (2)):P_F-CE_ = I_i_/(I_i_ + I_b_); P_CE-F_ = I_j_/(I_j_ + I_g_)(2)
where I_i_, I_j_, I_b_, I_g_ represent the integrated intensities of the resonance peaks of the F–P–CE, CE–P–F, F–P–F, CE–P–CE triads, respectively. Additionally, the average length of PF and PCE sequences in the copolymer are defined as:*L_PF_ = 1/*P_F-CE_; *L_PCE_* = 1/P_CE-F_(3)

The average PF and PCE sequence lengths and the corresponding degrees of randomness for the PPFPPCE block copolymers are reported in Table 1. Block length decreases as the reaction proceeds, while ***b*** is directly proportional to the mixing time. Therefore, it can be concluded that the experimental conditions we adopted allowed for the synthesis of several block copolymers, from very long blocks (PPFPPCE-5) to a random distribution (PPFPPCE-90), by simply changing the reaction time.

PPF film is more hydrophilic than PPCE, as expected on the basis of the electronegative oxygen atom present in the aromatic ring. As to the copolymers, the film hydrophilicity is correlated to the PF block length: the longer the PF sequences, the lower the water contact angle (WCA) value (Table 1).

The thermal degradation of the PPFPPCE-t is characterized by one-step weight loss that starts above 360 °C. PPCE is more thermally stable than PPF. This result can be explained considering that the presence of aliphatic rings imparts to polyesters a thermal stability higher than that of the corresponding terephthalic aromatic polymers, and that these last appeared to be more thermally stable than the furan-derived analogue [10,53]. The thermal stability of the copolymers is comprised between that of the two homopolymers.

The DSC curves and the corresponding main thermal transition data of the materials under study are reported in Figure 3 and Table 2, respectively. The two homopolymers display different phase behavior, as PPCE is semicrystalline, whereas PPF is completely amorphous, even though able to crystallize during heating once T_g_ is exceeded. Furthermore, T_m_,_PPF_ is 20 °C higher than that of PPCE. As to the glass transition, T_g,PPCE_ is below room temperature (12 °C), while the glass transition of PPF (56 °C) is higher than RT (Table 2), as expected on the basis of its aromatic nature. The copolymers are all semicrystalline, with the exception of PPFPPCE-90. The PPFPPCE-5 DSC trace shows two melting peaks, whose locations indicate the presence of the pure crystalline phases of PPF and PPCE. In the other copolymers, only one endothermic peak can be seen and, according to the semicrystalline nature of PPCE, it can be attributed to the PPCE sequences present in the copolymers. Those melting peaks move to lower temperatures and the heat of fusion decreases as the crystallizable block length decreases. The PPFPPCE-5 copolymer shows two glass transitions, indicating the presence of two amorphous phases: one PPCE-rich and the other PPF-rich. On the contrary, all the other copolymers are characterized by one glass transition in between those of the homopolymers, indicating that the two blocks turn miscible in the amorphous phase for L_block_ ≤ 4. PPFPPCE-5 and PPFPPCE-25 are able to crystallize above their corresponding T_g_s during heating, similarly to PPF. However, they are semicrystalline, being ΔH_c_ < ΔH_m._ Considering the ability of PPF to crystallize during heating as well as the cold-crystallization peak location, the exo-phenomenon can be attributed to the crystallization of the PPF sequences.

After melt quenching (Table 2 and Figure 3), the two homopolymers still display an opposite phase behavior. PPCE is semicrystalline, though able to crystallize on heating, whereas the PPF DSC trace exclusively shows the baseline deviation associated to the glass transition phenomenon. PPFPPCE-5 and PPFPPCE-25 are semicrystalline, the former being again characterized by the presence of two melting endotherms attributable to PPF and PPCE pure crystalline phases, while PPFPPCE-40 and PPFPPCE-90 are fully amorphous. As to the glass transition, an analogous behavior to the first scan can be observed. In conclusion, the crystallizing ability of PPCE is higher than that of PPF due to both the structural rigidity and the hindering of rotation of the furan ring [10,18]. For the copolymers, the ability to crystallize is lowered due to the shorter the length of PCE crystallizable sequences.

### 2.2. Mechanical Characterization

Tensile data (elastic modulus E, stress at break σ_b_, and elongation at break ε_b_) are reported in Table 3.

The two homopolymers are characterized by significantly different mechanical behaviors. E_PPCE_ is halved with respect to E_PPCE_, whereas ε_b,PPCE_ is 50 times higher, in accordance with a more flexible structure. The higher rigidity of the PPF can be explained taking into account that the furan ring-flipping is greatly suppressed [56]. The mechanical properties of the copolymers appear to be strictly related to the block length. In fact, the elastic modulus decreases and the elongation to break increases as the block length decreases. This trend is due to the different degree of crystallinity, which regularly decreased with crystallizable block length. Recently, Wang et al. investigated the mechanical properties of random poly(ethylene-co-1,4-cyclohexanedimethylene 2,5-furandicarboxylate) copolymers with different compositions. The authors found that the introduction of an aliphatic cyclohexane ring along the PEF polymeric chain improved the polymer toughness. In fact, the copolymer containing 60 mol% of the aliphatic ring was characterized by an ε_b_ of 186% [28]. In our case, the random copolymer (PPFPPCE-90) was characterized by an elongation to break of 635%, despite the lower amount of aliphatic rings present (50 mol% ca.), probably because of the introduction of a practically 100% *trans* aliphatic ring in the acid sub-unit.

### 2.3. Barrier Properties

Barrier properties exhibited by polymer films are an essential requirement for food packaging application. Food products need a complex type of protection in order to prolong their shelf-life, while maintaining the desired quality and characteristics. For example, living food-stuff that produce carbon dioxide (CO_2_) (such as fermented milk) require a packaging permeable to CO_2_ gas. At the same time, it has to be protected from oxygen (O_2_), which eventually oxidizes the fat. On the contrary, the packaging material of fresh vegetables and fruits has to be characterized by a moderate oxygen permeability. In fact, these foods are usually picked before ripening, therefore they need oxygen to respire. Animal-fat food reacts with oxygen, forming aldehydes that make the fat rancid, with consequent bad taste and flavor. Therefore, a high oxygen-barrier packaging material is required. Generally speaking, the shelf life of a food product is thus dependent on several factors, such as composition, time, temperature, gases, moisture, and light.

Taking into account the complex scenario reported above, the permeability behavior to three different gases, CO_2_, O_2_, and N_2_ was investigated. In Figure 4, gas transmission rate (GTR) values to CO_2_, O_2_, and N_2_ measured at 23 °C for PPF, PPCE, and PPFPPCE-t are reported. In all cases, CO_2_ gas was more permeable than O_2_ and N_2_, despite the larger molecular diameter (3.4 Å, 3.1 Å, and 2.0 Å, respectively) because of the diffusivity increase and solubility decrease with the size of the permeant [57]. It is worth highlighting that, as already observed for other furan-based polyesters [42,44], CO_2_ and O_2_ transmission rates were quite similar. This effect can be explained considering that the polar character of the furan ring causes a higher CO_2_ solubility in the polymer matrix.

PPF displayed significantly lower gas permeability than PPCE, in spite of the semicrystalline nature of the latter. This result is due to several factors including the hindering of ring flipping [10], limited subglass local dynamics conferred by the furan ring [56], and the establishment of C–H···O interactions among adjacent polymer chain segments [58]. Lastly, it should be noted that at room temperature, PPF was in the glassy state (Table 2). Therefore, macromolecular chains have little mobility, with a consequent reduction of the free volume for diffusion of the gas molecules through the polymer matrix.

PPFPPCE-t copolymers showed gas permeabilities comparable to those of PPF, notwithstanding the 1:1 molar composition. Specifically, the barrier properties of PPFPPCE-5 were slightly worse than those of PPF. This result can be explained based on the fact that for this polymer, due to the very high block length, both T_g_s and T_m_s of the two homopolymers were detected (Table 2). Therefore, preferential pathways for the gas crossing through interphase domains can be hypothesized. On the other hand, all other copolymers displayed a lowering of the gas permeability with respect to PPF. For PPFPPCE-25 and PPFPPCE-40, this phenomenon could be ascribed to the development of a certain degree of crystallinity, which hampers gas diffusion due to the restricted mobility of this phase, however, the same effect cannot be invoked for PPFPPCE-90, which is amorphous, with T_g_ very close to room temperature (Table 2). Nevertheless, it was possible to prepare nicely free-standing films. The results here reported may therefore suggest the development of unusual structural features in PPFPPCE-90, different from the classical crystalline phase, due to a synergistic effect of the furan ring, present in PPF, and of the cyclohexane ring of PPCE. Both rings can act as mesogenic groups and, in the presence of a flexible segment such as the propylene sub-unit, could contribute to the formation of 1D/2D ordered domains, typical of liquid crystal polymers, that, as is known, are characterized by outstanding barrier and mechanical properties, such as those found for PPFPPCE-90 [59]. A similar behavior has been already evidenced for an amorphous cyclohexane-containing copolymeric system [48]. Further studies are ongoing to investigate these unusual structural features. 

Temperature is one of the most important parameters affecting both food respiration rate and polymer gas permeability [57]. Therefore, to better understand the barrier behavior of PPFPPCE-t copolymers under different operating conditions, CO_2_ permeability has been measured in the range 8–40 °C. In particular, three different temperatures were chosen to contemplate all possible scenarios from food preservation to food handling. Furthermore, it should be noted that 8 °C was below T_g_ for all the investigated materials, while 40 °C was well above T_g_ (with the exception of PPF and PPFPPCE-5, this last displaying two T_g_s). Lastly, 23 °C was considered for copolymers with shorter blocks (i.e., PPFPPCE-25, PPFPPCE-40 and PPFPPCE-90) to be within the glass transition phenomenon.

In all cases, a clear dependence of the CO_2_-Transmission Rate (CO_2_-TR) with the temperature can be observed: the higher the temperature, the higher the permeability (Figure 5). However, depending on the molecular architecture, a different rate of increase can be observed. In particular, for PPF and PPFPPCE-5 the increase rate is constant because the tested temperatures are all below T_g_, thus, the polymers are in the glassy state. For PPFPPCE-25, a much higher increase can be noted from 23 (below T_g_) to 40 °C (above T_g_), due to the transition from a glassy to rubbery state. Finally, PPFPPCE-40 and PPFPPCE-90 displayed a more significant increment of CO_2_-TR from 8 to 23 °C than from 23 to 40 °C. This is because at 23 °C the transition glass-to-rubber is already happening. Of note, both PPFPPCE-40 and PPFPPCE-90 show lower permeability values than PPF in the whole range of investigated temperatures, and values below 0.04 cm^3^ cm m^−2^ d^−1^ bar^−1^ have been registered. This result is very important because it does not only certify the outstanding barrier properties of these copolyesters, but also testifies that they can be employed in a wide range of temperatures without suffering any significant performance decline. In particular, it can be noted that the high barrier performances are kept even above T_g_, and this can be considered as indirect evidence of the presence of the unusual structural phase mentioned above, which is responsible for the low permeability values to gases and remaining stable even at higher temperatures.

### 2.4. Film Color Determination

Envisioning possible applications as food packaging, film optical characteristics are of primary importance, as they impact on the consumer perception of the contained food. Therefore, films should be transparent and as colorless as possible. In this view, color and transparency of PPFPPCE-t films were studied and the results are reported in Table 4. As can be observed, PPCE displayed a higher tendency towards the white standard (lower ΔE) with respect to PPF and PPFPPCE-t copolymers. Furthermore, in terms of a* (red/green index) and b* (yellow/blue index), it can be noted that PPF and PPFPPCE-t showed a yellowish tendency (Figure 1), as confirmed by the h_ab_ values (slightly above 90°).

## 3. Materials and Methods 

### 3.1. Materials

The 2,5-Furandicarboxylic acid 98% (FDCA, CHEMOS GmbH & Co. K), *trans*-1,4-cyclohexane-dicarboxylic acid 97% (CHDA, TCI, Tokio, Japan)), 1,3-propanediol (PD), titanium tetrabutoxide (Ti(OBu)_4_), and titanium tetraisopropoxide (Ti(O-i-Pr)_4_) (Sigma-Aldrich, Saint Louis, MO, USA) were reagent grade products. FDCA, CHDA, PD, and Ti(O-i-Pr)_4_ were used as supplied, Ti(OBu)_4_ was distilled before use.

### 3.2. Synthesis of Homopolymers

Poly(propylene furanoate) (PPF) and poly(propylene cyclohexandicarboxylate) (PPCE) were synthesized in bulk by the usual two-stage melt polycondensation starting from 1,3-propanediol and the appropriate dicarboxylic acid, in accordance to the procedures previously reported [16,55]. 

### 3.3. Synthesis of Poly(Propylene 2,5 Furandicarboxylate/1,4-Cyclohexanedicarboxylate) Copolyesters

Poly(Propylene 2,5 Furandicarboxylate/1,4-Cyclohexanedicarboxylate) copolyesters were obtained by melt-mixing non-purified PPF and PPCE (1:1 molar ratio) in a 200 mL glass reactor at 240 °C under nitrogen atmosphere to prevent hydrolytic degradation. A head stirrer equipped with a teflon moon-shaped shaft was used. Mixing speed was set to 100 rpm. During the process, samples were taken from the reactor at different reaction times (5, 25, 40, and 90 min) and cooled in air. Copolymer formation was catalyzed by the residual Ti(OBu)_4_ catalyst introduced in the polymerization of PPCE.

### 3.4. Film Preparation

Films were obtained by compression molding in a Carver C12 laboratory press (Carver, Wabash, IN, USA) at a temperature equal to T_m_ + 30 °C for 3 min under a pressure of 2 tons·m^−2^. Prior to characterization, films were stored under room temperature for 4 weeks. The film thickness was measured with a DMG Sample Thickness tester (Brugger Freinmechanik GmbH, Munich, Germany).

### 3.5. Physicochemical Characterization

#### 3.5.1. Molecular Characterization

The polymer structure, composition, and sequence distribution were determined by ^1^H-NMR at room temperature (RT) on a Varian INOVA 400 MHz instrument (Palo Alto, CA, USA). The samples were dissolved (15 mg·mL^−1^) in a 20/80 (*v*/*v*) mixture of trifluoroacetic acid/chloroform-d solvent containing 0.03% (*v*/*v*) tetramethylsilane.

Molecular weight data were obtained by gel-permeation chromatography (GPC) at 30 °C using a 1100 Hewlett Packard system (Santa Clara, CA, USA) equipped with a PL gel 5 m MiniMIX-C column and a UV detector. A 95/5 *v*/*v* mixture of chloroform/1,1,1,3,3,3-hexafluoro-2-propanol was used as eluent with a 0.3 mL min^-1^ flow and sample concentrations of about 2 mg·mL^−1^. The system was calibrated with polystyrene standards in the molecular weight range 2000–100,000 g/mol.

#### 3.5.2. Thermal Characterization

TGA was carried out under nitrogen on a Perkin Elmer TGA7 (Perkin Elmer, Waltham, MA, USA). Gas flow of 30 mL·min^−1^ and heating scan of 10 °C·min^−1^ were used.

A Perkin Elmer DSC6 (Perkin Elmer, Waltham, MA, USA) was used for the calorimetric measurements. Aluminum pans containing about 10 mg of polymeric samples were heated up from −70 to 40 °C above melting at a rate of 20 °C·min^−1^ (I scan). A II scan was recorded under the same operating conditions after quenching to −70 °C.

#### 3.5.3. Mechanical Characterization

Tensile analysis was run on rectangular samples (50 × 5 mm) using a starting grip-to-grip separation equal to 23 mm and a crosshead speed of 50 mm/min. A Z2.5 Zwick Roell Texture machine (Ulm, Germany) equipped with a 500 N load cell was employed. The results are reported as the average of at least 5 replicates.

#### 3.5.4. Water Contact Angle Measurements

A KSV CAM101 instrument (KSV Instruments, Helsinki, Finland) was used for static contact angle measurements by analyzing the side profiles of 4 µL deionized water drops deposited on the film surface. The values reported are the average of at least 8 measurements.

#### 3.5.5. Permeability Characterization

Permeability tests were performed by a manometric method using a Permeance Testing Device, type GDP-C (Brugger Feinmechanik GmbH, Munich, Germany), according to ASTM 1434-82, DIN 53 536, ISO 15105-1, and to the method A of the Gas Permeability Testing Manual (Registergericht München HRB 77020, Brugger Feinmechanik GmbH). Measurements were carried out on samples of 78.5 cm^2^ at 8, 23, and 40 °C with a gas stream of 100 cm^3^·min^−1^, 0% RH. Chamber and sample temperatures were controlled by an external thermostat, KAAKE-Circulator DC10-K15 type (ThermoFisher Scientific, Waltham, MA, USA). O_2_, CO_2_, and N_2_ 100% pure food grade gases were used. All experiments were run in triplicate.

#### 3.5.6. Film Color Measurement

Film color and transparency were measured in accordance with ASTM E308 on a HunterLab ColorFlex EZ 45/0° spectrophotometer (Hunterlab, Reston, VA, USA). A calibration with black and white tile was run before the analysis. Results are reported as L* (lightness), a* (red/green), and b* (yellow/blue) parameters. The total color difference (ΔE) was calculated as follows:ΔE = [(ΔL)^2^ + (Δa)^2^ + (Δb)^2^ ]^0.5^(4)
where ΔL, Δa, and Δb represent the variation of each parameter with respect to a standard white plate used as background. Hue angle (*h*_ab_) was determined as follows [60,61]:*h*_ab_= tan^−1^ (b*/a*)(5)

Measurements were carried out in triplicate.

## 4. Conclusions

The results reported in the present work show that reactive blending is a very powerful tool to tailor the final properties of materials by simply acting on the mixing time. This parameter regulates the copolymers’ molecular architecture in an effective and economic way.

A strict correlation between block length and final properties was established (for a fixed molar composition). In particular, the smart properties of PPF were significantly improved by introducing the more flexible aliphatic cyclohexane-based sequences. As a matter of fact, the PPF thermal stability increased by about 10 °C. Moreover, the flexibility given by the PCE sequences had a remarkable effect on the mechanical and barrier properties of PPF. As to the tensile behavior, stiffness and deformability significantly varied depending on the molecular architecture. For the shortest blocks, the flexibility effect due to the PCE sequences drastically changed the PPF mechanical response. The elastic modulus decreased by six-fold, whereas the elongation to break increased by about 200-fold, overcoming the problem of the high rigidity of PPF.

Similarly, the permeability to CO_2_, O_2_, and N_2_ gases was remarkably influenced by the copolyesters’ molecular architecture. Of great importance is the significant improvement of the already good barrier performance of PPF for copolymers with short block length; these exceptional barrier properties were maintained even at high temperatures above material T_g_, allowing for their use in a wide range of temperatures without suffering any significant performance decline.

The introduction along the PPF macromolecular chain of more flexible PCE sequences seems to favor the development of an unusual structural phase characterized by 1D/2D order, which confers to these novel, 100% biobased materials the advantageous typical characteristics of liquid crystals.

## Figures and Tables

**Figure 1 ijms-20-02187-f001:**
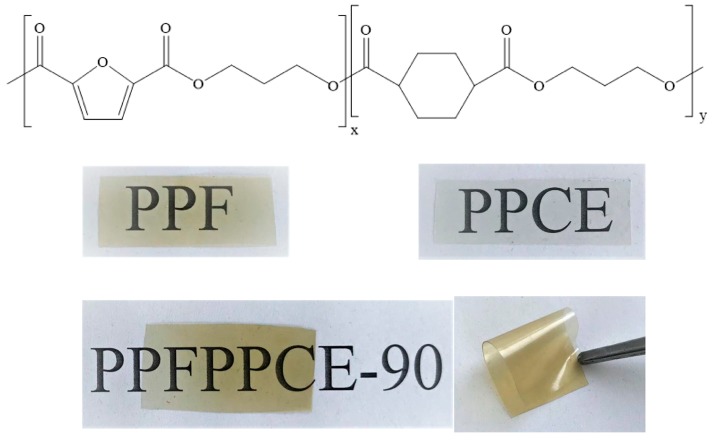
Chemical structure of PPFPPCE-t copolymers and pictures of compression-molded films of PPF, PPCE, and PPFPPCE-90.

**Figure 2 ijms-20-02187-f002:**
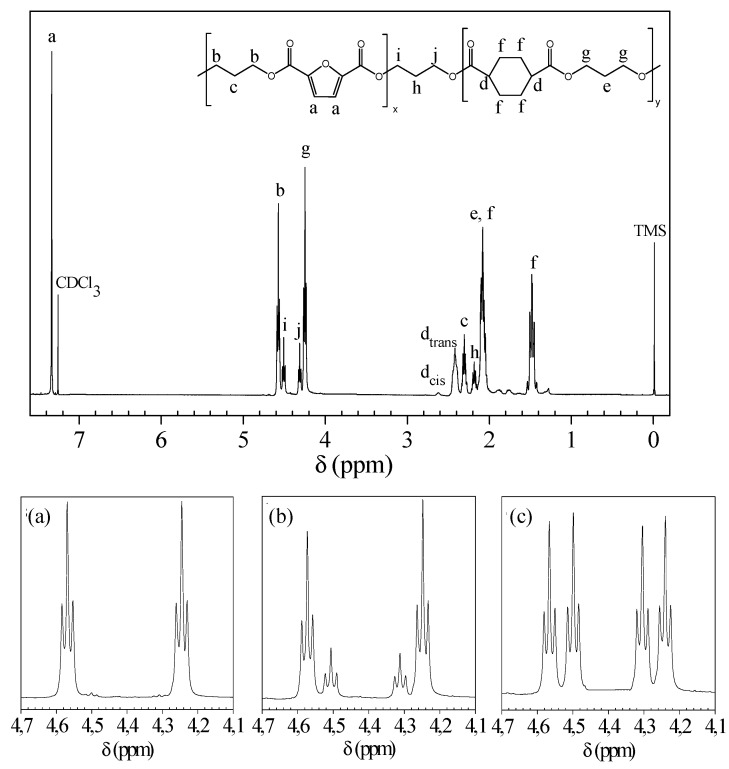
Top: ^1^H-NMR spectra of PPFPPCE-40 with resonance assignments; Bottom: Enlargement of ^1^H-NMR spectra in the region of 4.1–4.7 ppm. Evolution of the spectrum as a function of the mixing time: (a) PPFPPCE-5; (b) PPFPPCE-25; (c) PPFPPCE-90.

**Figure 3 ijms-20-02187-f003:**
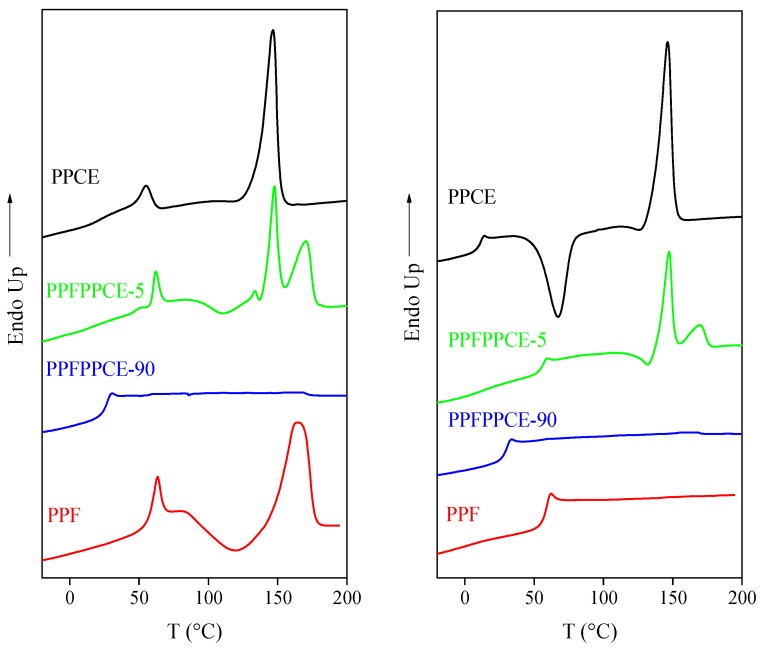
Calorimetric curves of PPF, PPCE, PPFPPCE-5, and PPFPPCE-90. Left: 1st scan; right: 2nd scan after quenching from the melt.

**Figure 4 ijms-20-02187-f004:**
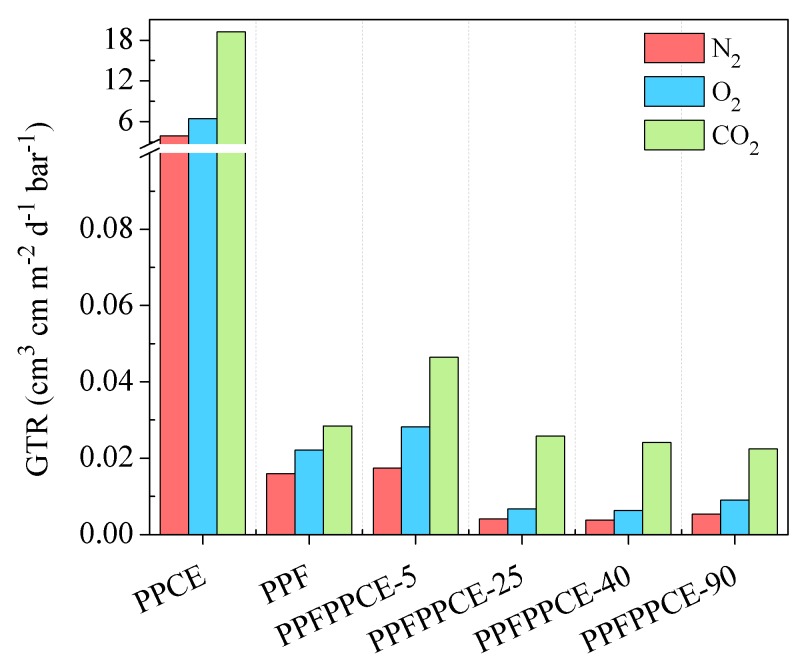
Gas Transmission Rate (GTR) of O_2_, N_2_, and CO_2_ through PPCE, PPF, and PPFPPCE-t copolymers.

**Figure 5 ijms-20-02187-f005:**
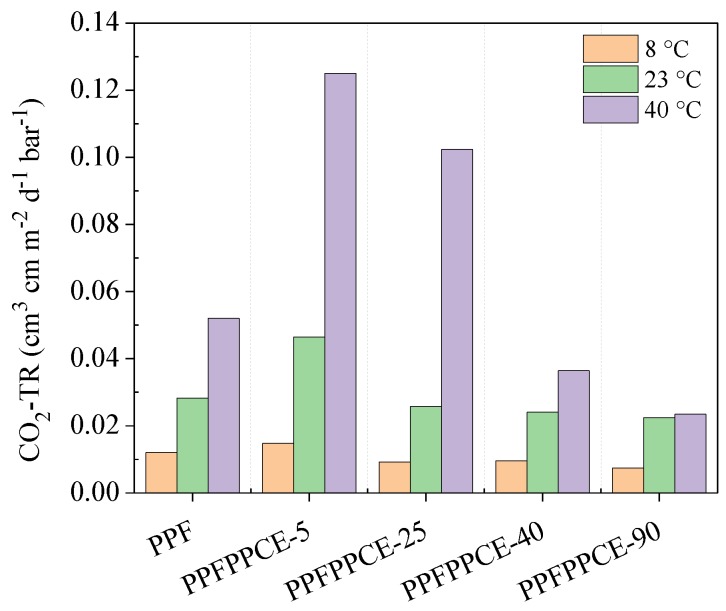
CO_2_-TR as a function of T (°C) for PPF and PPFPPCE-t copolymers.

**Table 1 ijms-20-02187-t001:** Molecular characterization and wettability data of PPF, PPCE, and PPFPPCE-t copolymers.

Polymer	*b*	L_PF_	L_PCE_	PF (mol%) by ^1^H-NMR	M_n_ (g/mol)	D	WCA (°)
PPF	-	-	-	100	30,000	2.3	88 ± 3
PPCE	-	-	-	0	33,000	2.5	97 ± 3
PPFPPCE-5	0.08	25	26	48	24,200	3.3	89 ± 2
PPFPPCE-25	0.45	4.2	4.8	47	25,100	2.6	91 ± 3
PPFPPCE-40	0.69	2.7	3.0	47	27,300	2.3	92 ± 4
PPFPPCE-90	1.00	2.0	2.1	49	28,400	2.3	94 ± 3

**b**: degree of randomness; **L_PF_**: average length of PF sequences; **L_PCE_**: average length of PCE sequences; **M_n_**: average number molecular weight; **D**: polydispersity index; **WCA**: water contact angle.

**Table 2 ijms-20-02187-t002:** Thermal characterization data of PPF, PPCE, and PPFPPCE-t copolymers.

			1st scan						2nd scan					
Polymer	T_id_ (°C)	T_max_ (°C)	T_g_ (°C)	ΔC_p_ (J/g°C)	T_c_ (°C)	ΔH_c_ (J/g)	T_m_ (°C)	ΔH_m_ (J/g)	T_g_ (°C)	ΔC_p_ (J/g°C)	T_c_ (°C)	ΔH_c_ (J/g)	T_m_ (°C)	ΔH_m_ (J/g)
PPF	360	387	50	0.19	137	7	168	7	50	0.19	-	-	-	-
PPCE	386	411	12	0.240	/	/	146	29	11	0.172	67	20	146	28
PPFPPCE-5	367	392	14; 55	0.064; 0.071	120	6	147; 167	16; 8	12; 55	0.07; 0.15	-	-	146; 169	8; 4
PPFPPCE-25	372	407	30	0.231	105	2	131	14	28	0.131	-	-	146	1
PPFPPCE-40	370	404	28	0.229	/	/	122	8	26	0.260	-	-	-	-
PPFPPCE-90	370	399	26	0.269	/	/	/	/	26	0.284	-	-	-	-

**T_id_**: degradation onset temperature; **T_max_**: temperature of the maximum weight loss rate; **T_g_**: glass transition temperature; **ΔC_p_**: heat capacity increment associated with glass-to-rubber transition; **T_c_**: crystallization temperature; **ΔH_c_**: heat of crystallization; **T_m_**: melting temperature; **ΔH_m_**: heat of fusion.

**Table 3 ijms-20-02187-t003:** Mechanical characterization data of PPF, PPCE, and PPFPPCE-t copolymers.

Polymer	E (MPa)	σ_b_ (MPa)	ε_b_ (%)
PPF	1363 ±158	31 ± 3	3 ± 1
PPCE	662 ± 52	16 ± 2	154 ± 18
PPFPPCE-5	1072 ± 52	27 ± 3	4 ± 1
PPFPPCE-25	951 ± 38	11 ± 1	28 ± 7
PPFPPCE-40	290 ± 51	8 ± 1	417 ± 82
PPFPPCE-90	228 ± 18	7 ± 1	635 ± 44

**E**: elastic modulus; **σ_b_**: stress at break; **ε_b_**: elongation at break.

**Table 4 ijms-20-02187-t004:** L*, a*, b*, ΔE, and h_ab_ of PPCE, PPF, and PPFPPCE-t copolymers.

Sample	L*	a*	b*	ΔE	*h* _ab_
White standard	66.80 ± 0.06	−0.72 ± 0.01	1.06 ± 0.06	-	124.2
PPCE	63.69 ± 0.39	−0.99 ± 0.04	2.49 ± 0.42	3.43	111.7
PPF	58.59 ± 0.20	−1.19 ± 0.06	15.42 ± 0.22	9.05	94.4
PPFPPCE-5	59.94 ± 0.27	−0.70 ± 0.05	12.68 ± 0.35	13.49	93.2
PPFPPCE-25	60.27 ± 0.80	−0.72 ± 0.15	10.12 ± 1.44	11.17	94.1
PPFPPCE-40	59.96 ± 0.16	−0.79 ± 0.07	10.85 ± 0.29	11.50	94.2
PPFPPCE-90	58.34 ± 0.35	−0.88 ± 0.06	10.90 ± 0.86	12.57	94.6

**L***: lightness; **a***: red/green index; **b***: yellow/blue index; **ΔE**: total color difference; ***h*****_ab_**: hue angle

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
