# Peer review of "Block Copolyesters Containing 2,5-Furan and trans-1,4-Cyclohexane Subunits with Outstanding Gas Barrier Properties"

_ijms, 2019, doi:10.3390/ijms20092187_

Round 1
Reviewer 1 Report
The manuscript describes furan and cyclohexane based block copolyester, biopolymers, with high gas barrier properties. The idea to produce block copolymers for sustainable applications from bio source is an important research topic. The obtained mechanical and gas barrier properties of these polymers are impressive. However, the experiments and manuscript writing can be much improved for publication.
As for the manuscript writing, too many typos (e.g. many double periods) throughout the manuscript could have been avoided. In the last paragraph of the introduction, the material design is given but the design principle is not clearly stated. More importantly, the manuscript is not prepared up to the standard for publication. For example, for all tables, the symbols should be noted, in addition to the explanation in the text. Another example, PPFPPCE-# is not noted anywhere in the manuscript. The manuscript is written with tedious descriptions of experiments rather than with concise and precise experimental results and discussion. In description of film color determination, no reasons are given for the necessity for these measurements.
As for the experiments, the provided NMR spectra and data shown in Table 1 do not provide persuasive evidences to prove either copolymer formation or physical mixing. For example, if PPFPPCE-5 is suggested to be with no significant chemical reactions, i.e. resembling physical mixing, then the molecular weight of PPFPPCE-5 should be the average of the two homopolymer. However, data in Table 1 do not support the suggestion.
As the gas barrier properties is the focus of the manuscript, its rationalization is of great importance and should be verified. However, rationalization is provided withour verification.
Author Response
Dear Reviewer 1,
thank you very much for your useful comments. We hope that our answers will be satisfying for you and that our manuscript was improved in scientific soundness as well as in presentation.
The manuscript describes furan and cyclohexane based block copolyester, biopolymers, with high gas barrier properties. The idea to produce block copolymers for sustainable applications from bio source is an important research topic. The obtained mechanical and gas barrier properties of these polymers are impressive. However, the experiments and manuscript writing can be much improved for publication.
1) As for the manuscript writing, too many typos (e.g. many double periods) throughout the manuscript could have been avoided.
We thank the reviewer for the suggestion. The manuscript has been thoroughly revised.
2) In the last paragraph of the introduction, the material design is given but the design principle is not clearly stated.
We thank the reviewer for the suggestion. The design principle has been added in the text (pag. 2 lines 37-40).
3) More importantly, the manuscript is not prepared up to the standard for publication. For example, for all tables, the symbols should be noted, in addition to the explanation in the text. Another example, PPFPPCE-# is not noted anywhere in the manuscript.
As requested, the symbols in the tables have been noted. On the other hand, the meaning of PPFPPCE-t was already indicated at the end of paragraph 3.3, where the synthesis of the copolymers is described. However, it has been shifted to the introduction section for sake of clarity.
4) The manuscript is written with tedious descriptions of experiments rather than with concise and precise experimental results and discussion. In description of film color determination, no reasons are given for the necessity for these measurements.
We thank the reviewer for the comment. As stated above, the manuscript has been throughout revised and its quality has been improved. Furthermore, an explanation on the importance of film color determination has been added to the text (see pg. 9 lines 20-23).
5) As for the experiments, the provided NMR spectra and data shown in Table 1 do not provide persuasive evidences to prove either copolymer formation or physical mixing. For example, if PPFPPCE-5 is suggested to be with no significant chemical reactions, i.e. resembling physical mixing, then the molecular weight of PPFPPCE-5 should be the average of the two homopolymer. However, data in Table 1 do not support the suggestion.
We thank the reviewer for giving us the opportunity to clarify this point. The reviewer is right. The term “physical blend” may be misleading, as the PPFPPCE-5 copolymer is not exactly a mere physical blend. In fact, although the block length is very high, transesterification reactions already occurred. This is testified by the presence of two additional triplets in the NMR spectrum, whose intensity is very low but not zero, thus indicating that the process just began.
In addition, it may be worth underlying that the process was carried out at high temperature (240°C) in order to promote the transesterification reactions. Such a high temperature wouldn’t be necessary if the final aim was to prepare a simple physical blend. Furthermore, as already observed in the literature, at the beginning of the reactive blending process a decrease of molecular weight is observed because chain scissions prevail over transesterification reactions. On the opposite, as the reaction proceeds, an increase of the molecular weight of the formed copolymers is recorded, as due to the prevalence of transesterifications over chain scissions. This is why the molecular weight of PPFPPCE-5 is not the average of the two homopolymers. Given all these considerations, the manuscript has been revised accordingly.
6) As the gas barrier properties is the focus of the manuscript, its rationalization is of great importance and should be verified. However, rationalization is provided without verification.
We thank the reviewer for the comment. As specified in the manuscript, the exceptionally low gas permeabilities of the PPFPPCE copolymers can be ascribed to the development of “unusual structural phase characterized by 1D/2D order”, also called mesophase. The formation of mesophase is favored in materials containing stiff segments (like furanic and cyclohexane ring) alternated with flexible ones (like aliphatic linear trimethylene subunit). Mesophases are typical of liquid crystals and are known to confer very high gas barrier properties to the materials.
However, the presence of these ordered domains is not easily detectable, especially without the aid of specific and less common analytical techniques. In this manuscript, we have focused on the synthesis and characterization of these materials, posing the attention on the effect of chemical architecture on the materials’ functional properties, i.e. tensile and gas barrier behavior. The study and characterization of the mesophase is ongoing and will be the object of a different work focused on this specific aspect.
Reviewer 2 Report
Dear Authors,
Below you will find my notes.
Lack of order: the second should be materials and methods, then results and conclusions
Introduction: Page 2, lines 13-14: these data come from 2001 and does not value nowadays
Figure 5: Can you translate please the Chinese signs in the legend?
Could you add pictures of obtained materials?
With the best wishes,
Reviewer
Author Response
Dear Reviewer 2,
thank you very much for the time spent in the revision of our manuscript.
We hope to well improved the quality and presentation, according to your suggestions.
1) Lack of order: the second should be materials and methods, then results and conclusions
We thank the reviewer for the comment. However, we followed the order indicated in the journal template, where materials and methods are reported after the results and discussion section.
2) Introduction: Page 2, lines 13-14: these data come from 2001 and does not value nowadays
We are sorry, but there is no reference from 2001. All the cited literature, with the exception of a book from 2006, is from 2012 or more recent (mostly from 2016-2018). Furthermore, the data on plastic packaging cited in the introduction are taken from a very recent publication of “PlasticsEurope”, which was made available online in the last months of 2018.
3) Figure 5: Can you translate please the Chinese signs in the legend?
We thank the reviewer for the comment. However, we do not see any chinese sign in the manuscript. This is most probably an artifact that occurs when creating the pdf file.
4) Could you add pictures of obtained materials?
We thank the reviewer for the comment. As requested, a picture of the obtained films has been added (see Figure 1 of the revised version).
With our best regards.
Round 2
Reviewer 1 Report
The revision is much improved though corrections are still needed. First of all, I agree with reviewer 2 that there are some strange characters (degree C in ) in the legend Figure 5 of the manuscript pdf file. As for all the temperatures shown in the manuscript, a space should be inserted between the number and the unit. In page 9, 'certifies' and 'testifies' are to be corrected.
The authors responded 'we have focused on the synthesis and characterization of these materials, posing the attention on the effect of chemical architecture on the materials’ functional properties, i.e. tensile and gas barrier behavior. The study and characterization of the mesophase is ongoing and will be the object of a different work focused on this specific aspect.' With these statements, then the manuscript title 'Outstanding gas barrier properties of 2,5-furan and trans-1,4-cyclohexane based copolyesters as a result of the synergistic effect of the two rings' shall be revised.
Author Response
Dear Reviser,
thank you very much for your comments.
We improved the manuscript following step by step your suggestion:
The revision is much improved though corrections are still needed
we perform another check of the whole manuscript, thank you.
1) First of all, I agree with reviewer 2 that there are some strange characters (°C) in the legend of Figure 5 of the manuscript pdf file.
We thank the reviewer for the comment. We have modified the figure and we think it’s ok now.
2) As for all the temperatures shown in the manuscript, a space should be inserted between the number and the unit.
We thank the reviewer for the comment. We have modified the manuscript as requested.
3) In page 9, 'certifies' and 'testifies' are to be corrected.
We thank the reviewer for the comment. Tha manuscript has been modified accordingly.
4) The authors responded 'we have focused on the synthesis and characterization of these materials, posing the attention on the effect of chemical architecture on the materials’ functional properties, i.e. tensile and gas barrier behavior. The study and characterization of the mesophase is ongoing and will be the object of a different work focused on this specific aspect.' With these statements, then the manuscript title 'Outstanding gas barrier properties of 2,5-furan and trans-1,4-cyclohexane based copolyesters as a result of the synergistic effect of the two rings' shall be revised.
We thank the reviewer. The title of the manuscript has been changed in:“Block copolyesters containing 2,5-furan and trans-1,4-cyclohexane subunits with outstanding gas barrier properties”.